# Breast-Feeding from an Evolutionary Perspective

**DOI:** 10.3390/healthcare9111458

**Published:** 2021-10-28

**Authors:** Juan Brines, Claude Billeaud

**Affiliations:** 1Department of Pediatrics, University of Valencia, 46010 Valencia, Spain; juan.brines@uv.es; 2Neonatology & Nutrition, CIC Pédiatrique 1401, INSERM, Hopital des Enfants, CHU Pellegrin Place Amelie Raba Leon, 33076 Bordeaux, France

**Keywords:** breastfeeding, history, infant, milk formulas

## Abstract

Lactation is the most critical period of mammal feeding given the compulsory dependence on milk of the offspring during a more or less extensive period following birth. This has also been the case for the human species until relatively recent times when heterologous milk processing has allowed the alternative of artificial lactation. The advantages and disadvantages of natural and artificial lactation (formula) have been widely discussed from the biological, psychological and cultural perspectives, without reaching a general agreement among the breastfeeding women themselves or among the health professionals concerned. On the subject of breastfeeding, the information available is enormous but as in other instances, the excess of it has often made it difficult to acquire objective knowledge on the matter that has hindered decision-making in specific circumstances. This situation is understandable given the diversity and the innumerable contingencies that the mother and health professionals must face in their natural and social (cultural) environments. To reduce these difficulties and taking into consideration the biological and cultural aspects involved in infant feeding, this article analyzes some aspects of the subject from the point of view of biological evolution as the mother-child dyad, mother-infant conflicts, in particular the conflict of weaning, late-onset primary lactase deficiency and the prevention from childhood of adult diseases. All of which allows to offer a testimony of gratitude and respect to women who have assumed the responsibility of breastfeeding their infants because without them the human species would not have existed.

## 1. Introduction

The feeding of the child, especially the infant, has been a matter of the utmost importance in every modern society. This attitude is fully consistent because without children no society has a future.

Feeding at the mother’s breast from birth to weaning is an obligatory requirement for all mammals for their survival. It has also been so for the human species until very recent times. Although society as a whole has been concerned in ensuring the feeding of children, there is no doubt that the human being most directly involved has been always the woman, especially those who breastfed [1].

But the issue of breast-feeding has exceeded the personal sphere of women involving national and supranational health institutions such as the WHO, which has repeatedly devoted considerable attention to it. An example is the categorical declaration of 1979:

“*Breast feeding is an integral part of the reproductive process, the natural and ideal way of feeding the infant and a unique biological and emotional basis for child development. This, together with its other important effects, on the prevention of infections, on the health and well-being of the mother, on child spacing, on family health, on family and national economics, and on food production, makes it a key aspect of self-reliance, primary health care and current development approaches. It is therefore a responsibility of society to promote breast-feeding and to protect pregnant and lactating mothers from any influences that could disrupt it*”[2]

This imposing statement could lead one to consider the superiority of breast-feeding in all circumstances, even the most extreme ones, thus ignoring possible and natural mother and infant exceptions. These contingencies, together with scientific and technical advances in nutrition, genetics, and psychology in a multiple and changing social environment, have promoted critical reviews that have eroded the foundations of its, in an earlier time, indisputable efficacy [3].

On the other hand, the incessant research activity from biological, psychological and cultural perspectives has provided an enormous amount of information about the mother, the child and the very diverse social environments in which the phenomenon of breast-feeding takes place. It is not strange the countless books, journals, and articles dedicated to the subject. But as in other instances, the excess of information has often made it difficult to acquire a general knowledge on the matter that has hindered decision-making in specific circumstances.

Before going into details, and in the search for solid foundations that can serve for general understanding and application it is appropriate to provide some principles:

The first one states that without reference to the past the present is incomprehensible and the future unpredictable. Every past happens to the present through a succession of events that we know as evolution.

The second argues that the two main approaches of the past from which we can understand the current actuality of breast-feeding are the biological and the cultural. The first one includes all the biological aspects of lactation as a peculiar phenomenon of a class of animals, that of mammals; the second one collects historical information on this phenomenon since the dawn of mankind.

The third postulate holds that the future of any animal community inescapably depends on the survival of its offspring; and limiting the claim to our species, our conviction that without the mother who has breastfed her infants, mankind would not exist [4].

## 2. Evolution of the Human Being

As we have advanced in the evolution of the human lineage, two simultaneous, intertwined and complementary processes can be distinguished: one biological (hominization) and the other cultural (humanization). *Hominization* is understood as the sequence of biological changes that, starting from the common hominid ancestor, has led to the current human species. *Humanization*, for its part, comprises the sequence of psychological and social changes that have shaped current human culture, differing it from those of the rest of the species. On the subject of infant feeding biological evolution has given rise to breast milk and the practice of breast-feeding; cultural evolution has led to artificial milk (milk formula) and the practice of bottle feeding.

### 2.1. Theory of the Evolution of Living Beings by Natural Selection

The number, type and diversity of living beings is countless and each one of them is, due to its characteristics, unique. The existence of more than five million species between known and still unknown is accepted but undoubtedly there were many more, probably hundreds of millions, most of them unknown [5].

The mechanism that Darwin proposed to explain evolution is *natural selection* by which organisms with heritable traits that favour survival and reproduction in a specific environmental setting will have more offspring than their less well-endowed counterparts [6], so the number of individuals with the phenotypic characteristics that facilitate their adaptation to the environment will increase over the subsequent generations. In this way, natural selection, determined by the environment, is a phenomenon that would act permanently, driving the sequence of adaptation of living beings to their environment [7]. This sequence is the result of a long evolutionary process that has spanned more than 3.500 million years [8] during which living matter has evolved in response to the changing diversity of environmental conditions [9] generating multiple species adapted to them.

The classical theory establishes, among others, the following statements:(1)The current beings descend from ancestors their lineages converging in a first hypothetic common ancestor (*Last Universal Common Ancestor—LUCA*).(2)Changes were gradual, offering a continuum of forms that happened imperceptibly with each generation from the depth of biological time.(3)The changes have been driven towards divergence which has generated multiple species.(4)The causes of these changes continue to operate today and therefore are susceptible to experimental analysis.(5)These changes are produced by the interaction of hereditary and environmental factors.

Thus, although this theory was initially based on the observation and classification of living beings, the subsequent discovery of the laws of heredity, the genetic analysis of populations, the advances in genetics, biochemistry and the advent of molecular biology have contributed to reach a coherent and systematic interpretation of the biological phenomena, not only of the organism as a whole but also of its components, functions, configuration and relationships. This intellectual construction, known as the *new synthesis*, *synthetic theory of evolution*, *modern evolutionary synthesis* or *neo-Darwinism* [7,10,11,12,13,14,15] constitutes a unitary conception of the processes of change in time of living beings by which large-scale evolutionary phenomena (*macroevolution*) are interpreted by inference from the genetic factors (*microevolution*) that determine them (mutations, genetic drift and migration). Modern synthesis has become the current paradigm of biology in the second half of the 20th century.

The role of genetics in the modern formulation of the theory of evolution has been decisive, placing it at the very core of evolutionary interpretation, since evolution establishes the development of dissimilarities between ancestral populations and their descendants [16].

The contribution of genetics to the knowledge of the evolutionary sequence has enriched and clarified some aspects of classical theory with some elementary notions [17]:(1)The unit of hereditary transmission is not the apparent character (*phenotype*) but the *gene*, the unit of innovation through the mechanism of mutation and meiotic recombination.(2)The unit of selection is the *individual* whose phenotype expresses the extremely complex interactions of the inherited genome and the environment.(3)The unit of evolution is neither the gene nor the individual but the *species* or rather, the Mendelian population that constantly shares, changes and recombines the gene pool made up from the set of genomes of all individuals.(4)The selection agent, the one that ultimately determines who will survive in order to procreate and thus transmit their favourable genes to offspring, is the *environment*. Environment will facilitate the resources of matter and energy with which each individual is modelled and by which the individual functions; it will also impose restrictions on survival and reproduction that will hinder or prevent the transmission of genes that express characters less adapted.

In any case, it is the mechanism of natural selection, acting on the variety of individuals, the engine of the dynamics of change and the one that decides the results of the survival of the fittest [14].

With this background and the contributions of physical anthropology and paleontology, among others, an agreement has been reached to consider the present human being as an animal included in the order of the *primates*, *hominid* family, genus *Homo* and species *sapiens*. We are *Homo sapiens sapiens*, closely related to gorillas, orangutans, chimpanzees and bonobos from whom we have differentiated as a result of an evolutionary sequence of millions of years. The fossil record and genetic studies support the idea that our origins took place from the common ancestor with the apes about six to eight million years ago [18], later diversifying into a relatively small number of related species that hybridized with each other [19]. Genetic and paleontological studies suggest a divergence of our line with Neanderthals and Denisovans more than half a million years ago. Anatomically modern humans appeared in Africa around 200,000 years ago with such a diversification that around 70,000 years ago there were up to six different species of humans [20]. *Homo sapiens sapiens* emerged in Africa as a unique species about 50,000 years ago and about 30,000 years ago human behaviour was basically modern, although since then until now, biological and, above all, cultural changes have not stopped taking place [21].

Although the theory of evolution constitutes one of the pillars of modern biology [22], it has not been immune to mainly religious but also scientific criticism. Even more, the study of the evolution of the human being, modern synthesis has focused on the biological processes of hominization, avoiding the concomitant cultural aspects.

### 2.2. Cultural Evolution

For more than a century the influence of the theory of evolution has far exceeded the limits of biology, becoming an integral part of the cultural equipment of Western civilization. The emergence and development of mankind are part of biological evolution in such a way that human beings cannot achieve a full and adequate understanding of themselves and their cultural development without the reference to their own biological background [16,23].

Both evolutions are related and have gone in parallel but at different rates: the first one continuously and slowly since the inception of life to the present, the second very quickly and recently since the appearance of civilization, and exponentially since the middle of the last century [23]. Cultural transmission may occur on short timescales, even within a single generation [24].

It should be highlighted that culture, understood as everything that is learned from others, which is transmitted repeatedly from one generation to another, forming accumulations of knowledge and practical skills common to specific social groups, is not an exclusive human phenomenon but rather similar activities have been observed in many vertebrates, mainly in apes, and even insects [25], but no living being depends as much on culture for its survival and reproduction than humans. In fact, the human being initiates social learning very early, easily assumes the opinions of others [26] and is a compulsive imitator [27].(Figure 1).

The changes of cultural evolution are acquired, not hereditary like those of biological evolution. Its acquisition takes place as a whole in a spontaneous or regulated way through social learning from multiple actors (parents, family, teachers, colleagues, the media, etc.) and various mechanisms [28].

This transmission of information includes knowledge, practical skills, and attitudes, and supposes the existence of a social memory that accumulates the knowledge of the past preserved in very different documentary supports (oral tradition, books, archives, etc.).

In addition, the changes are not based on the chance of mutations and natural selection, but are often directed to an end, that is, their genesis and evolution have a Lamarckian character (Transmissibility of the acquired modifications).

Despite their physical helplessness compared to other animals, cultural evolution has allowed human beings to adapt to their environment, an adaptation that has been carried out not by modifying their genome but changing their own environment.

The cultural heritage began in the form of guidelines for cooperation and mutual help between the members of the group, which promoted, as the communities became more extensive, the division of labour, the fundamental principle of articulation of complex societies.

### 2.3. Selective Factors of Cultural Evolution

Since the beginning of this century, it has been assumed that important factors in cultural evolution were a greater dependence on cooperation between members of the human group, social learning and cultural accumulation [18,23,25,29].

Group cooperation reduced the environmental risks, promoted the division of labour, improving the relationship between the benefit and the cost of human activity, and expanded access to resources.

The diffusion through learning and cooperation of beneficial innovations and their cultural accumulation made up for the biological deficiencies of the human being with respect to the external threats. Social learning and cooperation were fundamental activities for the evolution of *Homo sapiens* in the unstable environment of the Pleistocene [1,21].

## 3. Some Breast-Feeding Issues Considered from an Evolutionary Perspective

Sixty million years or more of mammalian evolution offers many opportunities for the biological study of lactation. The psychological and social conditioning factors add more complexity to the subject. Our claim will focus on four specific aspects that have aroused the interest of authors: Mother-child dyad, mother-infant conflicts, late-onset primary lactase deficiency and the prevention from childhood of adult diseases.

### 3.1. Mother-Child Dyad

The evidence that for millions of years breast-feeding has guaranteed the survival of hominids and among them, our own species, which has dominated the ecological niche, constitutes the strongest argument for its advantages, taking into account the absence of nutritional alternatives in ancient times.

But it does not follow from this evidence that natural selection has been operating to achieve an ideal nutritional product only for the infant, as is the conventional assumption, and this new attitude requires some additional comments:

The first is that the suitability of the breast-feeding process for the development and survival of the infant is the consequence of the selection, not the cause, since if the product were not proper the child would have died and the transmission of their genetic information would have disappeared to future generations. Conceiving the suitability of milk to meet the infant’s requirements on the wisdom and foresight of nature is a teleological illusion without scientific rigor. It is more convincing to resort to the action of natural selection that ensures the survival of only organisms adapted to environmental demands and resources.

The second comment that expands on the previous one is that the available evidence allows us to affirm without any doubt that selection works not by obtaining the best nutritional product for the child **but** by maximizing the survival of the mother-child pair, (mother-child dyad, in the words of Dugdale in 1986 [30]. And, it should be emphasized that, in evolutionary field, the survival of the mother is more important than that of the infant, given the immediate possibility of having more offspring. This evolutionary superiority, conditioned by its generative potential, is greater the younger the mother is.

This last statement deserves to be emphasized, since in professional milieu and in the food industries the goodness of the milk is judged solely, or mainly, based on the benefit to the infant and not based on the maximum benefit of the mother-infant dyad. And on this point, it should not be overlooked that the nutritional profit of breast-feeding for the child is comparable to the predation it imposes on the mother [31]. It is easily understood that both those genetic endowments that express an extremely insufficient synthesis of milk and those that synthesize it in excess lead to *the extinction of the lineage, the first one due to the death of the infant by starvation, the second due to the death of the mother by exhaustion.*

This rationale differs greatly from the conventional view that breast-feeding is ideal for the infant at all times and circumstances; and it also makes it possible to achieve some inferences about the volume and composition of woman’s milk. Indeed, it is clear that the total nutrients from breast-feeding must be, at the same time, sufficient to offer the child a relatively high probability of survival and the mother a minimum nutritional loss. In fact, it has been observed that in malnourished infants, mortality rate hardly increases until undernourishment is already noticeable [30]. Another directly related consequence is the benefit derived from reducing maternal overload by providing nutritional supplements to the child and to the mother herself.

The biological success of the human species, which has ruled the earth, allows us to infer that the maternal investment in raising infants has been the minimum to guarantee their survival and, at the same time, to avoid excessive investment; in this way a greater number of children will survive increasing the biological efficacy (*fitness*) of mother. In any case we must not underestimate the investment that women make in feeding their infant during breast-feeding, because strangely enough it is higher than that of pregnancy. Indeed, during the nine months of a normal pregnancy a new child is born with a weight of about 3.3 kg, most of which is water; nine months later the baby weighs 2.5 times more with a body composition poorer in water and richer in fat.

### 3.2. Mother-Infant Conflicts

The existence of conflicts between parents and children (*parent-offspring conflict*) is a very remarkable fact observed in many species of animals including mammals. They are usually conditioned by nutritional or sexual interests, or jealousy [1,32,33].

A very common form and possibly the one that manifests most precociously in our species is the *mother-infant conflict of weaning*. In primitive societies and in some traditional ones, events usually happen as follows: At the end of lactation, the volume of milk offered by the mother is insufficient to meet the needs of infant who has acquired a large amount of body mass. The latter tries by all means to maintain the easy feeding at breast, but the mother is already insufficient to supply the infant’s demand.

At the beginning of this period, the mother loses interest in breast-feeding and is less solicitous than usual about her infant’s wishes. The insistence of the infant ends up being annoying or painful so she reacts rejecting it. The child tries repeatedly to suckle and the mother becomes more and more energetic in her rejection. In this sequence of events, it is common for the infant to adopt regressive behaviours by showing whining, whimpering, or pretending to be sick. This situation is also observed in primates, being usually transitory and disappearing with integration in games with peers [23].

From classical evolutionary perspective, such conflict is explained by the confrontation of interests between the child and the mother. The first one tries to maximize the benefits of an easy and safe nutrition to the breast while the mother is preserved from an excessive biological wear and doing so she can have more children increasing so her biological efficacy.

### 3.3. Late-Onset Primary Lactase Deficiency (Adult Type)

Compared to other carbohydrates, lactose is an exceptional sugar in nature, where it is only found in small amounts, synthesized by some vegetables and fungi. Its role in biological evolution before the appearance of mammals must have been little or null since the magnitude of its synthesis compared to that of other carbohydrates such as fructose, sucrose, maltose, dextrins, starch, cellulose, etc., is insignificant.

Its link with mammals, its high concentration in milk and its generalization in them as a rapid energy source during lactation confers great interest to the study of its intestinal absorption and subsequent metabolism. The intestinal absorption is conditioned by lactase activity in the brush border of enterocyte that, as it is known, cleaves lactose molecule into glucose and galactose, both easily absorbable. And it is precisely in this aspect where it draws attention in a very significant way that, except for lactating mammals, its intake is harmful to the vast majority of animals causing meteorism, vomiting, diarrhea, vascular collapse, etc.; and, in addition, galactose, exhibits toxic properties, if ingested after lactation period, for a large number of metazoans as mammals, including humans.

From its origins, mammals incorporated the enzymatic endowment of intestinal lactase for the cleavage of lactose prior its absorption and its subsequent metabolization without risks for their creatures but restricted to the lactation period. This characteristic is also observed in the human species, but with peculiar features because, although the majority of adults are lactose intolerant, some ethnic and cultural groups ingest this sugar without inconvenience and show intestinal lactase activity, a fact that, since the last third of twentieth century has promoted a notable interest [34,35].

Most authors have considered lactase-persistent phenotype as a digestive adaptation and nutritional advantage, due to a cultural modification of the diet conditioned by the appearance of livestock about 8000 years ago [36] and the possibility to ingest, at any age, the milk of some mammals thus expanding food sources [21,37,38,39,40,41,42,43,44]. This view is reflected in a quote from Kretchmer [38]: “*If in particular populations it became advantageous to be able to digest milk, the consequent survival of individuals with a genetic mutation that leads to greater intestinal lactation activity in adulthood should be favoured. An individual who obtained this ability to digest lactose through this classical form of Darwinian adaptation would be expected to be able to transmit this trait genetically* “. This gene-culture coevolutionary hypothesis continues to be widely accepted [21,40,45].

Nonetheless the idea that the maintenance of lactase activity in the adult of our species constitutes a selective advantage seems weakly founded if the phenomenon as a whole is analyzed from an evolutionary perspective [46]. This new interpretation is based on several facts:

First, the quantitative representation of natural and sexual selection is biological efficacy (fitness) that assesses the reproductive success of individuals whose features have a selective advantage. Comparing the impact of such a supposed adaptive advantage of populations whose adults maintain intestinal lactase activity to those in which this activity is absent it does not seem any selective advantage since the people whose adults show intestinal lactase activity (Northern Europeans, Anglo-Saxons, a few African tribes of shepherds) have a much smaller population than that of intolerant ones (Chinese, Japanese, Indians, etc.).

Second, the duration of the period of heterologous milk consumption by the adult has been evolutionarily very short, especially for processes such as the one discussed in which the difference in biological efficacy is minimal or null, namely, there is no distinct mortality, and therefore there could not have been a differential selection of genes. But in addition, although the discovery of agriculture and livestock as a means of subsistence has about 8.000 years in the Middle East, its generalization to the rest of world was slow. On the other hand, the storage of dairy products for consumption frequently implied the loss of lactose substratum (cheeses, acidified milk, etc.) with which it could not act in the hypothetic way that has been supposed.

Moreover, the risk of adult to die from lack of milk is nil, while in the infant, at least in primitive societies, the lack of milk leads inexorably to death. In addition, the volume of milk in our species is not enough to meet the adult needs as it is for the infant who, unfortunately, lacks an alternative.

Finally, at a time when lactase-persistent phenotype was analysed in depth, studies in the adult Japanese population showed an incidence of milk intolerance about 19% [47], which is inexplicable in a population in which the historical consumption of milk and dairy products was practically non-existent before WWII. In this study lactase activity was significantly greater in milk drinkers than in non-drinkers and internationally, the activity was higher in those communities whose milk consumption was greater. These findings together with the demonstration that lactase activity can be experimentally induced in animals led the authors to consider that environmental factors play a more important role than genetic factors in the pathogenesis of milk intolerance.

Other authors, however, that have tried to induce intestinal lactase expression with different lactose feeding protocols have shown lack of enzyme induction and consider that the lactose tolerance observed experimentally in animals and in human trials administering progressive doses of lactose is due to the growth of lactose-digesting bacteria in the colon (colonic adaptation), which enhances colonic lactose processing and possibly results in the reduction of intolerance symptoms [45].

So the adult tolerance to lactose is an issue still active but for the reasons given and for other series of considerations that exceed the limits of this article, we support the idea that the disappearance or attenuation of intestinal lactation activity that is observed in all mammals and, in our species, after the first years of life, far from being an inconvenience, it constitutes a selective advantage since, due to its toxicity for most adults, breast milk, and to some extent fresh heterologous milk, is safeguarded for infants.

It therefore seems clear that the loss of intestinal lactase function after weaning becomes a selective advantage for young mammals allowing the continuity of the lineage [48]. The cases in which this activity remains beyond weaning, would be explained more by the maintenance of the enzymatic induction of intestinal lactase due to the uninterrupted supply of lactose than by resorting to evolutionary interpretations.

### 3.4. Prevention in Childhood of Adult Diseases

The main health problems in developed countries, except times such as the current COVID-19 pandemic, are the so-called diseases of civilization or non-communicable diseases (NCD). These conditions include nutritional, metabolic, and endocrine diseases (overweight, obesity, type 2 diabetes mellitus and osteoporosis), hypertension, atherosclerosis and cardiovascular consequences (angina pectoris, myocardial infarction) and stroke. Other NCD are cancer (lung, breast, skin, bladder, colon), chronic pulmonary disease, allergic diseases and asthma.

The importance of them is out of the question: In the early years of the 20th century one in five humans died from cardiovascular disease and cancer, today this figure amounted to four out of five.

NCD account for about 41 million deaths per year (71% of the world’s annual deaths), 15 million of which are premature deaths (<69 yrs). Cardiovascular diseases, cancer and chronic respiratory diseases are responsible for around 40%, 20% and 10% respectively of these deaths [49].

A relevant fact since the middle of the last century has been the worldwide decline in breast-feeding, a phenomenon that has coincided with the increase in the frequency of NCD. The following question immediately arises: Is there a causal relationship between the decrease in breast-feeding and increase of NCD? The benefits of breast-feeding for the mother health are undeniable, calculating that the globalization of such practice could prevent 20.000 annual deaths from breast cancer [50]. But for the infant and its future as an adult, the answer is not so clear [51]

Some authors affirm that breast-feeding has beneficial effects for life [9,50]; others are not so convinced. From the perspective of biological evolution, which has often been offered to defend the benefit of breast-feeding, it is difficult to give a satisfactory answer. We must remember that the very dynamics of natural selection only allow us to ensure the advantages of breast-feeding for the interval from birth to the end of the effective reproductive period, a period that in the remote past would not last more than 30 years [52]. Beyond this age, it is unlikely that such advantages were expressed since their features were not subjected to the sieve of natural selection. From an evolutionary perspective a prerequisite for a genetic trait to be advantageous is that it must be passed on to offspring, that is, it must be hereditary; any other character that, while offering advantage to its bearer, cannot be transmitted to the offspring lacks evolutionary interest and disappears with the individual. At this point it must be emphasized that we are referring exclusively to the transmission of characters encoded by genetic information; cultural transmission, the fastest and most effective mechanism of change in human history, is largely unrelated to these considerations.

Most of NCD present curiously their statistical mode, after the usual age of childbearing; they appear in a period of the life cycle in which the selection pressure does not exist or is much weaker. As we know, the transmission possibilities of mutations present in advanced ages, “genetic garbage” in Medawar expression [53], are almost anecdotal.

The childbearing age, therefore, marks the upper limit of the period to which the benefits of breast-feeding could extend in the shaping of the human species, since over time those populations with the best natural milk were selected to the detriment of those who did not have enough milk or it was of poor quality.

These thoughts, as can easily be understood, are speculative, and, in some instances, more than one explanation has been offered. It is clear that to get more satisfactory answers to the possible influence of breast-feeding on the genesis of NCD we need systematic study based on a blend of man’s ancient biological mammalian heritage, modern scientific knowledge (mainly long-term and wide-ranging epidemiological and clinical investigations), and awareness of cultural successful traditional time-tested adaptations.

Thus, the current health situation in high-income countries is such that many of the contingencies for which breast-feeding offered advantages in the past have disappeared due to the control that mankind exercises over the environment as well as technical and healthcare advances. If the hypothesis that the child’s diet can condition the health of adults is valid, acting during critical periods of development, the possibility of carrying out nutritional interventions from infancy to reduce the impact of NCD is open [54,55].

## 4. Conclusions

This contribution has tried to cover several objectives. The first of them has been to offer a testimony of gratitude and respect to the women who have borne the responsibility of breast-feeding their infants, because without them mankind would not have been possible. Immediately we are obliged to issue a message of reassurance to those women who for biological, psychological or social reasons have not been able to breastfeed because scientific, technological and industrial advances have achieved such a degree of perfection in alternative dietary products to woman’s milk that feeding the child with them is safe. Finally, to regret the decision of those women who have decided not to breastfeed and have resorted to artificial lactation for aesthetic reasons, for fashion, selfishness or comfort, since they have been deprived of a unique and enriching life experience and of a motive to express their generosity towards the being that is more his own, his son. These reflections are not trying to condemn their attitude because being men and having known during decades of professional exercise many of the sociocultural contingencies that continue to hinder breastfeeding, any reproach would be, a priori, a daring.

## Figures and Tables

**Figure 1 healthcare-09-01458-f001:**
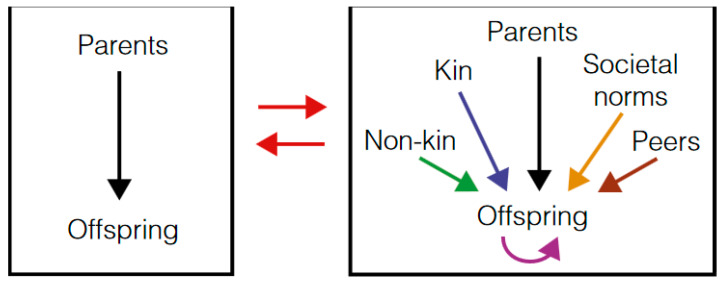
The factors involved in the transmission of cultural evolution (learning) are more complex than those of biological evolution (genetic inheritance) and can occur in short time intervals even in a single generation (modified from Creanza, Kolodny and Feldman).

## Data Availability

Not applicable.

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
