# Peer review of "Breast-Feeding from an Evolutionary Perspective"

_healthcare, 2021, doi:10.3390/healthcare9111458_

Round 1

Reviewer 1 Report

I found the paper interesting as it gives a different perspective about breastfeeding, focusing on more ancestral and evolutionary issues. As it is a perspective, I do not feel the need to change the narrative or to propose significant changes.   

Author Response

This article is a evolutionary process of breastfeeding and the originality is difficult to judge as an usual article 

it’s an historical article a perspective on the breastfeeding evolution  I appreciate the revision oof reviewer 1

and I hope that it’s will be need  just one day English révision 

Reviewer 2 Report

This very well-written article offers an interesting perspective on breast-feeding and its evolution that is worth sharing and reading.

Some considerations:

Breast-feeding has - in my opinion - two different major components that have implications on evolution, consequences for the childs short- and long-term outcomes and overall health. One is the act of breastfeeding itself (implications for mother-infant bonding, possible influence on short- and longterm psychological health of mother and infant) and the other is the exposure to human milk (source of pre- and probiotics with a strong influence on the microbiome development which in turn is associated with multiple longterm health outcomes, including NCD). I would like to hear the authors optinion on if the article should elucidate these different components throughout the article, or emphazize more clearly what component is talked about in which paragraph. Maybe also a clarification in the introduction would be useful?

Infroduction: Annother/ a next "level" in the history of breast-feeding and human milk are the recent approaches to "instutrialize" mothers abilities to produce human milk by processing human milk in a way that different components of the milk (f.e. proteins) can be extracted and then given as supplements to preterm infants (or very sick term infants) because it is hypothesized that human supplements are beneficial as when compared to animal-derived supplements. This of course has huge ethical implications. This could (not must) be an interesting addition to the story.

401/402: "others are not so convinced" (reference please).

I dont know if it can be scientifically doubted that breast-feeding has beneficial effects for life and health as an adult. Also: are we talking about the act of breastfeeding or  just human milk? Both have shown to have beneficial short- and long-term effects for the infant. This becomes particularly clear when looking at studies that have been performed with preterm infant cohorts that frequently cannot be breastfed and/or cannot receive human milk. The beneficial effetcs become particularly clear looking at these studies. Can you clearifiy?

30 Would recommend to not use the exact same wording in Abstract and introduction. (line 30)

285: why son´s not infant`s?

Author Response

Breast- Feeding from an Evolutionary Perspective. 211010

REVIEWER 2. 

Thank you very much for your kind comments and suggestions.

Sincerely yours.

  1. Brines and C.Billeaud.

Consequently, the authors feel obliged to add some details to facilitate understanding of the objectives, structure and content of the text.

From the outset, it is essential to state that this is not a common research paper with its Introduction, Material and Methods, Results, Conclusions, Comments and Bibliographic References, but rather that it is an updated review on breastfeeding, from the dual perspective of biological and cultural evolution. The main objective of this paper was to serve as a scientific and cultural frame of reference for the Congress "The first 1000 days of life", organized by the (European Association for Pediatric Education (EAPE/AEEP) and the Human Milk Foundation, in which infant feeding, especially human milk, was a central theme. The meeting was designed particularly for physicians, mainly pediatricians and obstetricians, and nurses, particularly midwives and pediatric nurses.

It is not, therefore, a simple disclosure of the subject but a serious updating of it from the two most solid scientific perspectives for professionals, already mentioned but taking care of the language so that the principles could be clearly perceived by any health agent. Therefore, it does not require the methodology of a clinical trial or research work.

In any case, there are conclusions. They are so obvious that they do not need investigation. No mammal is known that has been able to survive after birth without the milk of its species, mainly that of the mother, and, in some cases, that of another surrogate mother or nurse. And the same happened in humans until the domestication of some species of mammals (goat, sheep, cow, camel) allowed the alternative of heterologous milk feeding. And it is worth remembering that these antecedents go back to the Mesopotamian populations about 8,000 years ago, which represents a short period of time since the appearance of mammals, about 65 million years ago so it is a phenomenon reduced in time. and in space.

The first comment of the reviewer is very interesting and it is worth taking into account in further insights into the matter. There is no doubt that at present it is convenient to distinguish between the effects of the act of feeding the infant to the breast with the mother's whole milk, and the current possibility of using various individualized components of human milk for prophylactic and therapeutic purposes.

Breastfeeding has had the main advantages of reducing infant mortality, promoting intimate contact with the mother directly that facilitates mental development and the first social integration with her (promotion of bonding) and with relatives, mainly through mother. On the other hand, the possibility of using whole human milk (milk banks) or some of its individualized components has opened up new prophylactic and therapeutic possibilities, especially in the prematurely born infant. There is no doubt about the progress of these technical innovations with the spread of milk banks in the feeding of premature babies and, specifically, in the prevention of necrotizing enterocolitis.

But in both cases we have not a sufficient time perspective to apply the principles of biological evolution. In these cases, the methodology that the authors describe would be applicable to know the impact of breastfeeding on conditions whose mode is beyond the reproductive life period, that is, to resort to systematic studies  based on a blend of man's ancient biological mammalian heritage, modern scientific knowledge (mainly long-term and wide-ranging epidemiological and clinical investigations), and awareness of cultural successful traditional time-tested adaptations .

Row 401/202:  Bibliographic references:

Hytten F., Science and lactation. In: Falkner F (ed). Infant and child nutrition worlwide. Issues and perspectives. Boca Raton. CRC Press. 1991: 117-140.

Brines J. Colloquium. En: Battaglia F, Falkner F, Garza C, et al. (eds). Maternal and extrauterine nutritional factors. Their Influence on fetal and infant growth. Salamanca: Ergon, 1996: 456-57.

            Regarding the possible benefits of breastfeeding on adult life the very dynamics of natural selection only assure the advantages of breast-feeding for the interval from birth to the end of the effective reproductive period, a period that in the remote past would not last more than 30 years. Beyond this age, it is unlikely that such advantages were expressed since their genetic endowment was not transmitted. Any other character, even beneficial to the individual, if not transmitted (not hereditary), to the offspring lacks evolutionary interest and disappears with the individual. Cultural transmission, the fastest and most effective mechanism of change in human history, is largely unrelated to these considerations.

            But it is pertinent to provide an additional comment. Overweight and obesity are more frequent in children who are fed with artificial lactation than at the breast. Consequently, the possibility of beneficial effects of breastfeeding on non-communicable diseases of the post-reproductive adult linked to obesity (hypertension, aterosclerosis, cardiovascular diseases and stroke) could be speculated.

Row 30. The paragraph on line 30 could be changed to the following: “Feeding at the mother's breast from birth to weaning is an obligatory requirement for all mammals for their survival”.

Row 285. We agree: We change “son” for “infant”.

Reviewer 3 Report

Manuscript details: 
Journal: Healthcare 
Manuscript ID: healthcare-1402609 
Type of manuscript: Perspective 
Title: Breast- Feeding from an Evolutionary Perspective  

Special Issue: The First 1000 Days of Infant
Authors: Juan Solanes Brines, Claude Billeaud
Submitted to section: Perinatal and Neonatal Medicine,

Hi,

I want to thank the authors for the article entitled “Breast- Feeding from an Evolutionary Perspective.” The article is well written, looks scientifically sound and backed up by many references, and can be accepted for publication.

Regards

Author Response

Breast- Feeding from an Evolutionary Perspective. 211010

REVIEWER 3. 

Thank you very much for your kind review.

Sincerely yours.

  1. Brines and C. Billeaud

October the 10th, 2021.

Reviewer 4 Report

This manuscript provides a unique view into the the evolution of breast-feeding. It is a delight to read as a popular science paper with many interesting references for humanity. The subject is important but the format of the manuscript is in my opinion not within the scope of this scientific journal. If the manuscript was submitted as a review article with a major revision of the text and references, the manuscript would be within the scope of MDPI Healthcare. To give one example: Are breastfeeding beneficial for children? On row 402 only two references are given to answer this and they are both almost 50 years old. The whole 3.4 section needs to be revised and inspiration for this can be found in the already mentioned reference 49.

The aim of the paper needs to be clearly stated. Methods are not mentioned at all. Conclusions are drawn without reference to aim/objectives.

Some other suggestions:

  1. Row 4/5 - add addresses to authors
  2. Row 39 - include citation marks "" for this citation
  3. Row 71 - why "my"? there are two authors.
  4. Row 96 not 3500 billion years. It should be 3500 million years.
  5. Row 166 - why specify "Western"? Isn't civilisation enough?
  6. Row 286/286 - why has the child a sex in the text? 
  7. Row 381 - reference to Covid situation might be a big thing at the time of publishing but feels really outdated when readers see this text in a couple of years. 
  8. Row 391-394 - Reference needed
  9. Reference 53 looks very interesting by the title but could not be retrieved from Google Scholar or pubmed, please add doi. 
  10. Motivate why Reference 54 is relevant for this publication, it appears to be inappropriate self-citation. 

Author Response

Breast- Feeding from an Evolutionary Perspective. 211010

REVIEWER 4. 

We sincerely appreciate your detailed and accurate review.

Sincerely yours.

  1. Brines and C.Billeaud.

In relation to the reviewer's first comment, the authors consider it convenient to provide some considerations that will facilitate understanding the structure, content and objectives of the paper.

From the outset it is necessary to state that it is not a common research article with its Introduction, Material and Methods, Results, Conclusions, Comments and Bibliographic References, but rather that it is an updated review on breastfeeding, from the dual perspective of biological and cultural evolution. The main objective of this paper was to serve as a scientific and cultural frame of reference for the Congress "The first 1000 days of life", organized by the (European Association for Pediatric Education (EAPE/AEEP) and the Human Milk Foundation, in which infant feeding, especially human milk, was the central topic. The meeting was designed particularly for physicians, mainly pediatricians and obstetricians, and nurses, particularly midwives and pediatric nurses.

It is not, therefore, a simple disclosure of the subject but a sound updating of it from the two most solid scientific perspectives for professionals, already mentioned but taking care of the language so that the principles could be clearly perceived by any health agent. Therefore, it does not require the methodology of a clinical trial or research work.

In any case, there are conclusions. They are so obvious that they do not need further research. No mammal is known that has been able to survive after birth without the milk of its species, mainly that of the mother, and, in some cases, that of another surrogate mother or nurse. And the same happened in human beings until the domestication of some species of mammals (goat, sheep, cow, camel) allowed the alternative of heterologous milk feeding . And it is worth remembering that these antecedents go back to the Mesopotamian populations about 8,000 years ago, which represents a short period of time since the appearance of mammals, about 65 million years ago, so it is a phenomenon reduced in time. and in space.

In this first comment, the reviewer literally expresses the following observations:

“To give one example: Are breastfeeding beneficial for children? On row 402 only two references are given to answer this and they are both almost 50 years old. The whole 3.4 section needs to be revised and inspiration for this can be found in the already mentioned reference 49.”

These reviewer lines skew and distort the actual information in the paragraph that refers to the extreme benefits of breastfeeding for the mother and infant. We transcribe the entire paragraph:

“The benefits of breast-feeding for the mother health are undeniable, calculating that the globalization of such practice could prevent 20.000 annual deaths from breast cancer (49). But for the infant and its future as an adult, the answer is not so clear. Some authors affirm that breast-feeding has beneficial effects for life (9, 50); others are not so convinced.”

Regarding the fact that only two authors from about fifty years ago (9, 50) are cited, it is due to the fact that at that time there were few researchers who clearly expressed their doubts regarding whether breast milk could be the best for the infant at all times and circumstances, controversy that has not yet ended. This current situation is accurately reflected in the article by Victora et al (2016) that the reviewer advises us to read. In the first two lines of the aforementioned article (49 of our text) it is literally said: “The importance of breastfeeding in low-income and middle-income countries is well recognised, but less consensus exists about its importance in high-income countries”, statement that in our opinion exactly matches our position on the matter.

What's more, does the reviewer really believe that the benefits for the infant of breastfeeding, which is by far the most generalized position of pediatricians, midwives and nurses, as well as of the most prestigious profesional institutions concerned (WHO, UNICEF, American Academy of Pediatrics, National Pediatric Societies, etc.), need  bibliographic support to sustain it it? Is there any need for evidence to affirm that the offspring of the genus Homo, which initiates the biological sequence that leads to modern human beings, like those of all mammals, depend after birth on breastfeeding from their mothers to survive?

But in addition, the reviewer, for reasons that are not made explicit, omits in his question the existence of authors who are currently not convinced that breastfeeding is the best option in all possible contingencies faced by the mother-child dyad, opinion which is considered heterodox for the majority of professionals involved in the subject of breastfeeding. This is a current reality, a source of strong controversies within pediatrics, a reality so clear that, curiously, it is collected in the first two lines of the Victora et al paper that the author recommends us to follow for section 3.4.

The reviewer advises that we be inspired to improve section 3.4 in the aforementioned work by Victora et al (2016). We do not think it is convenient at all. And not because we disqualify the work of Victora et al, which seems excellent to us, but because our approach to the problem of whether or not breastfeeding can influence the morbidity and mortality of NEC is evolutionary, a perspective absent in the aforementioned paper. And in this sense, it is worth insisting, that our position is that the very dynamics of natural selection only assure the advantages of breast-feeding for the interval from birth to the end of the effective reproductive period, a period that in the remote past would not last more than 30 years. Beyond this age, it is unlikely that any advantages were expressed since their genetic endowment was not transmitted. Any other character, even beneficial to the human being, if not transmitted (not hereditary), to the offspring, lacks evolutionary interest and disappears with the individual. Cultural transmission, the fastest and most effective mechanism of change in human history, is largely unrelated to these considerations.

But it is pertinent to provide an additional comment. Overweight and obesity are more frequent in children who are fed with artificial lactation than at the breast. Consequently, the possibility of beneficial effects of breastfeeding on non-communicable diseases (NCDs) of the post-reproductive adult linked to obesity (hypertension, aterosclerosis, cardiovascular diseases and stroke) could be speculated.

Some other suggestions

  1. Row 4/5 - add addresses to authors. OK
  2. Brines. Honorary Professor of Pediatrics. Departement of Pediatrics, Obstetrics and Gynecology. University of Valencia. 17, Av. Blasco Ibañez, 46010 Valencia. Former President of European Association for Pediatric Education. E-mail: “[email protected]

  1. Billeaud. President of European Association for Pediatric Education.

  1. Row 39 - include citation marks "" for this citation. OK.

“Breast feeding is an integral part of the reproductive process, the natural and ideal way of feeding the infant and a unique biological and emotional basis for child development. This, together with its other important effects, on the prevention of in-fections, on the health and well-being of the mother, on child spacing, on family health, on family and national economics, and on food production, makes it a key aspect of self-reliance, primary health care and current development approaches. It is therefore a responsibility of society to promote breast-feeding and to protect pregnant and lac-tating mothers from any influences that could disrupt it” (2).

  1. Row 71 - why "my"? there are two authors. OK.

            Corrected text:

            “The third postulate holds that the future of any animal community inescapably depends on the survival of its offspring; and limiting the claim to our species, our conviction that without the mother who has breastfed her infants, mankind would not exist (4).”

  1. Row 96. OK.

Original text:

“… is the result of a long evolutionary process that has spanned more than 3.500 billion years.”

            Incomprehensible and unforgivable mistake. The available evidence that is reproduced in any book on biological evolution is that life appeared on the face of the Earth about 3.500 million years ago.

Thank you very much for the correction.     

Corrected text :

“… is the result of a long evolutionary process that has spanned more than 3.500 million years.”

  1. 5. Row 166 - why specify "Western"? Isn't civilisation enough?

            We have used the term Western civilization to refer to the dominant one in the technologically developed countries that roughly corresponds to the high-income countries. In any case, the authors limited themselves to reproducing the terms used in the original article.

            We agree with the reviewer on the question of whether the Western designation is the best but the choice of another term would start a discussion that goes beyond the objective of the paper. It is possible that removing the term Western would imply a generalization of the lifestyle that would not reflect current reality.

  1. Row 286/286 - why has the child a sex in the text? OK.

Original text:

            “annoying or painful so she reacts rejecting him. The child tries repeatedly to suckle and”

            The error is due to a direct translation from Spanish

Corrected text:

“…annoying or painful so she reacts rejecting it. The child tries repeatedly to suckle and…”

  1. Row 381 - reference to Covid situation might be a big thing at the time of publishing but feels really outdated when readers see this text in a couple of years.

            It is possible that the COVID-19 pandemic will be dominated in two years but the authors have serious doubts that it can be forgotten. In any case, the reviewer will agree that it has been one of the worst pandemics that have affected humanity and that it will endure in the annals of medicine and history as the medieval black plague or, in the last century the flu or the poliomyelitis. It should be noted that without reference to the past the present reality is scientifically and culturally incomprehensible.

In any case, the purpose of the authors was to take as a reference the universal concern for the current COVID pandemic and compare it in importance with (NCD).

  1. Row 391-394 - Reference needed.

            Original text:

“NCD account for about 36 million deaths per year (about two-thirds of the world's annual deaths), 10 million of which are premature deaths (<60 yrs).  Cardiovascular diseases, cancer and chronic respiratory diseases are responsible for around 30%, 13 % and 7 % respectively of these deaths.”

This information has been obtained from the statistical figures that WHO publishes regularly and corresponds to previous years. The most updated that we have are dated April 21, 202. (https://www.who.int/news-room/fact-sheets/detail/noncommunicable-diseases).

The updated data and percentages would be those included in the corrected text.

Corrected text:

“NCD account for about 41 million deaths per year (71 % of the world's annual deaths), 15 million of which are premature deaths (<69 yrs). Cardiovascular diseases, cancer and chronic respiratory diseases are responsible for around 40%, 20 % and 10 % respectively of these deaths.”

  1. Reference 53 looks very interesting by the title but could not be retrieved from Google Scholar or pubmed, please add doi. OK.

            Requested reference:

“ISBN of the book: 84-86754-93-3. “

“Legal deposit f the book: M-44400-1996.”

  1. Motivate why Reference 54 is relevant for this publication, it appears to be inappropriate self-citation.

            Infant nutrition has been a topic previously and repeatedly addressed by the authors. The two main approaches to the subject are biological and cultural.

Biological evolution does not allow to ensure or deny whether breastfeeding can have an effect on NCDs, since its advantages or disadvantages take place in the reproductive period where there is a possibility of genetic transmission, which does not occur in the post-reproductive period where the majority of NCDs occur. This is not the case from the cultural perspective where nutrition and lifestyle interventions in school have proven decisive effectiveness. In this area, education, understood as the transmission of knowledge, practical skills and attitudes, is particularly relevant. And in the general educational space (family, media, peers, school) it is that of schooling where cultural transmission is institutionalized and carried out more effectively.

Linking the first 1,000 years of life with the pre-school and school period means taking advantage of the continuity of the child's development to deepen the possibilities offered by school in promoting a healthy life and in the prevention of adult diseases, mainly the NCDs.

            In any case, the suppression of this quote would subtract information that we consider important about the authors' educational experiences in the school setting, but would not alter the main line of the article.

Round 2

Reviewer 2 Report

I find the article very interesting for the community and very well-written. Since the authors very well adressed my comments I accept the article in the present form. If the article is published as orginal research /original articel or rather as a review is also up the journals editors but I appreciate the authors scientific approach and I would support the publication in the current form.

Reviewer 4 Report

thanks for valuable updates.